# Conditional Diffusion on Web-Scale Image Pairs leads to Diverse Image Variations

## Abstract

Generating image variations, where a model produces variations of an input image while preserving the semantic context has gained increasing attention. Current image variation techniques involve adapting a text-to-image model to reconstruct an input image conditioned on the same image. We first demonstrate that a diffusion model trained to reconstruct an input image from frozen embeddings, can reconstruct the image with minor variations. Second, inspired by how text-to-image models learn from web-scale text-image pairs, we explore a new pretraining strategy to generate image variations using a large collection of image pairs. Our diffusion model *Semantica* receives a random (encoded) image from a webpage as conditional input and denoises another noisy random image from the same webpage. We carefully examine various design choices for the image encoder, given its crucial role in extracting relevant context from the input image. Once trained, *Semantica* can adaptively generate new images from a dataset by simply using images from that dataset as input. Finally, we identify limitations in standard image consistency metrics for evaluating image variations and propose alternative metrics based on few-shot generation.

## 1 Introduction

Machine learning initially focused on optimizing and improving models on small datasets. The field has transitioned to training general purpose models on web-scale data and then finetuning them for specific tasks on smaller datasets. This paradigm shift has lead to state-of-the-art results on a number of different domains. In this paper, we focus on the relatively underexplored task of adapting an image generative model to different datasets. One approach is to simply train a generative model on a large dataset of unlabelled images and finetune them on smaller datasets. While this approach is straight-forward in theory, it requires clever architecture or regularizer design to prevent overfitting in practice (See Sec.2.2). As models scale up, finetuning for every dataset also just becomes increasingly impractical.

Image-conditioned diffusion models are now increasingly used to adapt generative models to new datasets, also known as *image variations* (Ye et al., 2023; Pinkney, 2022; Xu et al., 2023b). First, an image encoder trained on a different upstream task, such as self-supervised learning (DINO (Oquab et al., 2023)) or contrastive-learning on web-image text pairs (CLIP (Radford et al., 2021)) produces frozen embeddings. The frozen embeddings then condition a diffusion model usually pretrained on text-to-image, which is finetuned to reconstruct the original image. However in these models, the study of *image variations* often remains a secondary objective. In this paper, we take a step back and directly analyze image variations in isolation. To avoid ambiguity involved in generating multiple objects in an image, we focus our evaluation on datasets with a single dominant object. We start with a vanilla image-conditioned diffusion architecture that is composed of a frozen image encoder and conditions the diffusion model with cross-attention. As done in prior works, we train the diffusion model to reconstruct images from frozen embeddings. We demonstrate that without text-to-image pretraining or co-training, this model qualitatively achieves near-perfect image reconstruction. This suggests that the capacity to generate image variations via reconstruction is due to the implicit regularization inherent in the pre-trained or co-trained text-to-image model. While empirically this may be sufficient to generate plausible image variations, the relationship between the text-to-image model, image variations and scale remains unclear.

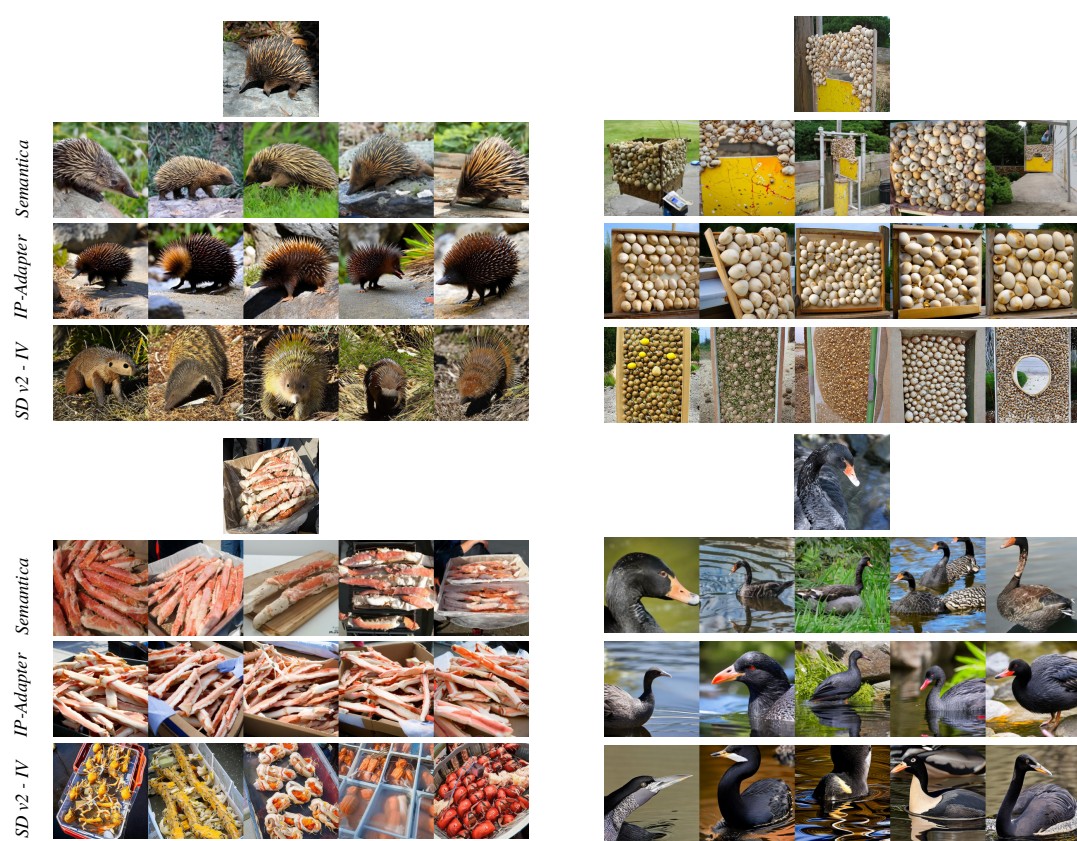

Figure 1: Each grid presents a conditioning image at the top followed by $512 \times 512$ image variations generated by Semantica, IP-Adapter, and SDv2 IV. Samples generated by a semantic image-variation model should maintain semantic consistency with the conditioning image while also being sufficiently diverse. Semantica demonstrates greater diversity than IP-Adapter while preserving semantic context. While SD v2 generates diverse outputs, the generated outputs are often not congruent with the context image. Additional samples are present in App. A.

In this paper, we explore a different pretraining strategy with the same image-conditioned diffusion architecture. We train our model *Semantica* using *image episodes*, which are image pairs that belong to the same web page. Therefore, training relies exclusively on the hypothesis that images from the same web page should have some common semantic attributes. For example, it is probable that images scraped from a Wikipedia page on dolphins, contain pictures of dolphins. To generate image variations, the model receives an image and then generates another image that preserves semantic information. Under this pre-training strategy, our experiments demonstrate that scaling both the image encoder and the diffusion decoder steadily improve image variation quality. In Fig. 1, we compare *Semantica* to state-of-the-art image variation models. *Semantica* is capable of generating high quality and diverse images, reflective of semantic information from the conditioning image.

Evaluating image variations is non-trivial. Unlike standard image generative modeling where a model generates images from scratch, a model has access to the entire test set of images via conditioning when generating image variations. This means a model could simply copy the conditioning image and achieve high scores both at distribution-level and instance-level metrics. To bridge this limitation in existing metrics, we instead propose to evaluate image-variations exclusively in the few-shot setting. Concretely, we limit the number of conditioning images available to the model and then measure its ability to model the test distribution.

Our main contributions are:

- Current techniques train diffusion models on image reconstruction to produce image variations. Our analysis shows that diffusion models trained to reconstruct images from frozen embeddings produce only minor low-level variations.

- We explore an alternative pretraining strategy for generating image variations. This involves conditioning a diffusion model with a random image from a webpage and training it to denoise a different random image from the same webpage.

- We rigorously compare DINOv2 and CLIP as frozen image encoders to produce image variations. While CLIP is now the standard image encoder, our experiments demonstrates that DINOv2 yields superior performance.

- Standard image-level metrics such as LPIPS and distribution-level metrics such as FID fail to capture diversity in image variations. To address this, we introduce few-shot metrics designed to assess the diversity in image variations.

## 2  RELATED WORK

### 2.1  IMAGE VARIATIONS

SD-V2 Image Variations (Pinkney, 2022), IP-Adapter (Ye et al., 2023), MultiFusion (Bellagente et al., 2024) and Verstatile Diffusion (Xu et al., 2023b) generate image variations via image reconstruction. Specifically, SD-V2, IP-Adapter and MultiFusion adapt a pretrained Stable Diffusion model. SD-V2 swaps the frozen CLIP text embedding with the CLIP image embedding, first only finetunes the cross-attention layers of the SD model to attend to the image embedding and then the entire backbone. IP-Adapter trains a adapter layer to the output of the clip image embedding and additional decoupled cross-attention layers. MultiFusion finetunes a LLM to accept additional image inputs. The resultant LLM embeddings than condition a pretrained Stable-Diffusion model. Versatile Diffusion trains a single model to perform both text-to-image and image variations, with some decoupled components such as cross-attention. All these methods use the same input image as the target image, and rely on regularization for variations. Bordes et al. (2021) analyze the reconstructions generated by diffusion models conditioned on just the global embedding from self-supervised models and show that they can reconstruct image semantics. Here we show, with cross-attention based conditioning, near perfect reconstruction can be achieved.

### 2.2  GENERATIVE TRANSFER

Prior to image variations, there has been research that studies the adaptation of source-pretrained generative models to a target dataset with adaptation of weights. Initial works study the transfer of discriminators and generators in GANs from a source dataset to a target dataset (Wang et al., 2018). Further, Grigoryev et al. (2022) show that ImageNet pretraining on a large GAN model is beneficial for transfer to small datasets. A number of papers focus on improving generation quality by adapting only a subset of parameters. These include scale and shift parameters (Noguchi & Harada, 2019), updating only the higher discriminator layers (Mo et al., 2020), linear combinations of scale and shift parameters (Shahbazi et al., 2021), modulating kernels or convolutions (Zhao et al., 2022a; 2020; Cong et al., 2020; Alanov et al., 2022) and singular values (Robb et al., 2020), mapping networks from noise to latents (Wang et al., 2020; Mondal et al., 2023; Yang et al., 2023) and latent offsets (Duan et al., 2024). Various works apply regularization losses by enforcing constraints to samples/weights by the source generator including elastic weight regularization (Li et al., 2020), domain correspondence (Ojha et al., 2021; Gou et al., 2023; Hou et al., 2022), contrastive learning (Zhao et al., 2022b), spatial alignment (Xiao et al., 2022), inversion (Wu et al., 2022; Kato et al., 2023; Thopalli et al., 2023), random masks on discriminators (Zhu et al., 2022) and alignment free spatial correlation (Moon et al., 2023). Given the increasing popularity of VQ-VAE and diffusion based models, recent work (Sohn et al., 2023) and (Zhu et al., 2022) explore few-shot finetuning on VQ-VAE tokens and diffusion models. We defer to Abdollahzadeh et al. (2023) for a detailed exposition of all these methods. In contrast to these works, we explore training a generator on web-scale images and study their transfer to standard small-scale image datasets. Retrieval augmented models (Casanova et al., 2021; Blattmann et al., 2022) compute nearest neighbours for a query image across a bank of memory images. These retrieved neighbors facilitate the training or generation

process. Unlike these methods, we do not require access to a memory bank of images during train or test time.

## 2.3 DIFFUSION

Diffusion and score-based generative models have become increasingly successful in modelling images (Ho et al., 2022b; Saharia et al., 2022; Nichol et al., 2022; Balaji et al., 2022), videos (Ho et al., 2022a; Singer et al., 2022) and audio (Kong et al., 2020). As generation quality has steadily improved, they have been used in contexts with more and more conditioning variables. Well-known examples are text-to-image and text-to-video modelling, where the conditioning variable is text. In this case, the conditioning variable can be seen as a sequence from which cross-attention layers communicate to the feature maps of the image or video, i.e. what the diffusion model is learning to generate (Saharia et al., 2022; Nichol et al., 2022). As the desire for controllable generation increases, solutions such as ControlNet (Zhang et al., 2023) have been developed. ControlNet takes in conditioning images of the same size as the generations, and uses a copy of the UNet to learn an encoder for the conditioning signals. Although this encoder trains fast due to parameter initialization from a pretrained diffusion UNet, it is difficult to deal with different sized inputs. In those cases, only conditioning via cross-attention as done in (Xu et al., 2023a) overcomes the in-place additions between the ControlNet encoder and the base UNet. Conditioning on images as context has produced impressive results, turning scribbles or edge detections into high quality image generations (Wang et al., 2023; Najdenkoska et al., 2023) and discriminative tasks (Bai et al., 2023; Li et al., 2023). In contrast with the above mentioned techniques, our framework relies on general web-based pretraining for semantic-based adaptive image generation. While (Giannone et al., 2022) study the transfer of few-shot diffusion models between small datasets (CIFAR-100 $\rightarrow$ miniImageNet, we see in Sec. 8.3 that this can lead to sub-optimal transfer. (Liu et al., 2023) employ test-time guidance using similarity scores with a reference image, to steer unconditional generative models. However, they still require training a separate unconditional generative model for each domain.

## 3 MODEL

### 3.1 DIFFUSION

Diffusion models learn to generate examples by gradually denoising a diffusion process. For a single datapoint, their loss can be expressed as a squared error between the original datapoint and its prediction:

$$\mathbb{E}_{t \sim \mathcal{U}(0,1), \boldsymbol{\epsilon} \sim \mathcal{N}(0,\mathbf{I})} \left[ w(t) || \boldsymbol{x} - f(\boldsymbol{z}_t, t, t) ||^2 \right] \text{ where } \boldsymbol{z}_t = \alpha_t \boldsymbol{x} + \sigma_t \boldsymbol{\epsilon}_t \tag{1}$$

It is helpful to define $\text{SNR}(t) = \alpha_t^2 / \sigma_t^2$. In the case of $w(t) = \text{SNR}(t)$ the loss above is equivalent to a loss in $\boldsymbol{\epsilon}$-space, the simple loss from Ho et al. (2020). After training, the denoising model generates samples by taking small steps. Starting at $t = 1$ with initial Gaussian noise and one slowly denoises for timesteps $t = 1, 1 - 1/N, \ldots$ where the number of sampling steps is $N$. Although many samplers are possible, in this paper we use the standard DDPM sampler (Ho et al., 2020).

### 3.2 IMAGE ENCODER

Training an image-conditioned diffusion model requires an image encoder that extracts semantic information from a conditioning image. We could train a separate ViT end-to-end as an image encoder with the diffusion model to learn useful conditioning representations. Instead, we leverage pre-trained image encoders and condition our diffusion model on their "frozen" representations. This offers two advantages. First, we can precompute representations for all images in the dataset that eliminates expensive forward passes through the encoder during training. Second, we can use different scales of readily available pre-trained encoders and just focus on scaling the diffusion model. We investigate ViT image encoders trained with two pretraining strategies, contrastive learning (SigLIP) (Zhai et al., 2023) and self-supervised learning (DINOv2) (Caron et al., 2021; Oquab et al., 2023).

## 3.3 DIFFUSION DECODER

In early days, diffusion literature typically used UNets that consisted of ResNet blocks, with optional self-attention layers. More recent architecture either use full Transformers (DiT (Peebles & Xie, 2023), StableDiffusion) or UNeTs with transformer backbones (UViTs) ((Hoogeboom et al., 2023). The transformer backbone makes it especially easy to use conditioning signals in these architecture via cross-attention layers. To be precise, the denoising neural network takes in a noised image at a certain timestep $z_t \in \mathbb{R}^{H \times W \times 3}$, timestep $t \in \mathbb{R}$ and contextual information $c \in \mathbb{R}^{T_c \times D_c}$. In principle it does not matter which diffusion or generative model we use to generate images. In practice we use the simple diffusion framework (Hoogeboom et al., 2023) because it can learn to generate high resolution images end-to-end without the need of a separate autoencoder.

## 3.4 CONDITIONING INFORMATION

Formally, the image encoder encodes the context image $X_c \in \mathbb{R}^{H \times W \times C}$ into a sequence of tokens $X_C \in \mathbb{R}^{T_c \times D_c}$, We follow the encoder-decoder framework in the original Transformer (Vaswani, 2017) to condition the diffusion decoder with context tokens. We employ conditioning only in the lowest-resolution transformer in UViT. Every self-attention block in the transformer backbone is followed by a cross-attention block, where the diffusion decoder cross-attends to $X_c$. In addition to conditioning via cross-attention, we also explore conditioning only using global features using the CLS token. Specifically, we normalize the CLS token, embed it with a dense projection and add it to the timescale embedding. The resultant embedding then conditions the diffusion model via FiLM layers (Perez et al., 2018).

## 4 IMAGE VARIATIONS VIA RECONSTRUCTION

A common technique to generate image variations is to incorporate image-specific context into the denoising objective using frozen embeddings.

$$\mathbb{E}_{t \sim \mathcal{U}(0,1), \epsilon \sim \mathcal{N}(0,\mathbf{I})} \left[ ||\boldsymbol{x} - f(\boldsymbol{z}_t, t, \boldsymbol{x}_c)||^2 \right] \tag{2}$$

where $f$ is a pre-trained model on a different objective, for example text-to-image modeling (Ye et al., 2023; Bellagente et al., 2024), $x$ is an image and $x_c$ are frozen embeddings from the same image.

The objective amounts to reconstruction where the diffusion model is trained to reconstruct the context image in pixel space from frozen embeddings. The generation of diverse image variations via reconstruction can be attributed to two primary factors.

1. The frozen embeddings from the image encoder retain information only useful for the upstream task it is trained on. This in turn, leads to a lossy representation of the original image and the diffusion model has to fill in the missing details.

2. The pretrained diffusion decoder provides implicit regularization that prevents the model from simply collapsing to the conditional input image.

We train a diffusion model from scratch to optimize Eq 2 with DINOv2 frozen embeddings. Qualitatively, we see that extremely early in training, less than 100K steps, the generated samples almost collapse to the conditional image. Fig. 2 displays three samples from the trained diffusion model, showing some very minor low-level variations but no high-level variations. Similar results can be seen using SigLIP frozen embeddings in App. B. This suggests that pretraining or jointly training on a text-to-image pretraining objective may be the principal source of image variations. While this may be sufficient to generate reasonable image variations, it is non-trivial to predict the relationship between $f$ and the quality of image variations. For example, does a bigger text-to-image model lead to better image variations?

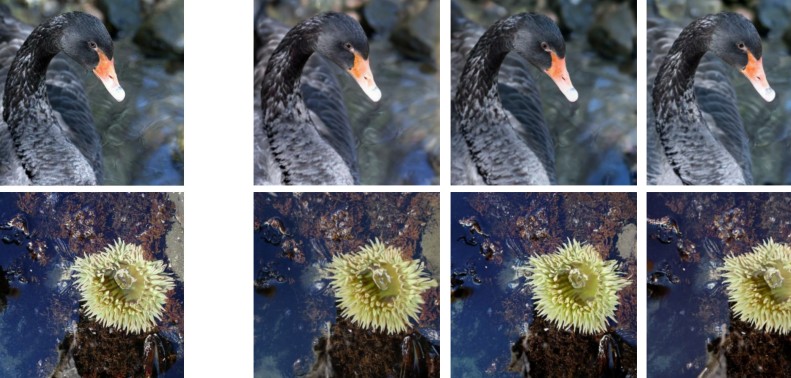

Figure 2: A conditional diffusion model reconstructs images from frozen DINOv2 embeddings. **Left:** Input Images. **Right:** Three samples from the trained diffusion model with guidance 0.0 exhibiting low-level variation but lacking high-level variation.

## 5    IMAGE VARIATIONS VIA WEB-SCALE IMAGE PAIRS

Inspired by web-scale image-text pretraining (Radford et al., 2021), we modify the denoising objective as follows:

$$\mathbb{E}_{t\sim\mathcal{U}(0,1),\epsilon\sim\mathcal{N}(0,\mathbf{I})}\left[||\boldsymbol{x} - f(\boldsymbol{z}_t, t, \boldsymbol{y}_c)||^2\right] \tag{3}$$

where $\boldsymbol{x}$ is an image from a webpage and $\boldsymbol{y}_c$ are frozen embeddings obtained from another random image from the same webpage.

In particular, we use Episodic WebLI (Chen et al., 2023), where each episode contains randomly sampled loosely related images (i.e., they are clustered according to their URL). *Note that Episodic Webli is explicitly deduplicated from all standard image train and test benchmarks.* While Episodic Webli was originally designed for training few-shot vision language models, we introduce a novel application by utilizing it to train image variation models. We randomly sample an image as the conditioning input $\boldsymbol{x}$ and another image from the same episode as the ground-truth "target" image $\boldsymbol{y}_c$.

Each episode consists of images that are loosely related, whereas our model assumes conditioning and target images share semantic information. This mismatch may lead the model to waste capacity on modeling irrelevant noisy conditioning-target pairs. To address this we filter out pairs with low similarity as done in image-text pretraining. The pretrained encoder computes the global CLS representation from the conditioning and target image. We compute the cosine similarity between the global conditioning/target representations and filter out pairs below a lower threshold. Unlike image-text pretraining, we also filter conditioning-target pairs with similarity above a high threshold. in order to ensure generation of interesting images. This can ensure that generated images retain the semantics of the conditioning image, while being sufficiently different and interesting. App. D provide more information on how the high and low thresholds were set. After filtering, we obtain a dataset of around 50M image pairs. Notably, despite utilizing a dataset an order of magnitude smaller than standard text-to-image datasets (Schuhmann et al., 2021), we achieve strong results on image variations.

## 6    EXPERIMENTS

### 6.1    ARCHITECTURE DETAILS

For our baseline *Semantica* model, we inherit all hyper-parameters from the ImageNet label conditioned diffusion model. The denoising model follows a U-ViT architecture that operates on $256\times256$ images. The architecture consists of a initial $1 \times 1$ convolution. The model has four downsampling stages, where each stage downsamples the feature maps by a factor of 2 at its output and a final

|          | SiD B | SiD L |
|----------|-------|-------|
| DINOv2 B | 13.0  | 9.8   |
| DINOv2 L | 11.7  | 9.0   |

|          | SiD B | SiD L |
|----------|-------|-------|
| SigLIP B | 17.1  | 12.9  |
| SigLIP L | 15.6  | 11.2  |

Table 1: We report the FID on ImageNet across two encoders (DINOv2 and SigLIP), two diffusion model sizes (SiD B and SiD L) and two encoder sizes (B and L) at 300K steps. DINOv2 encoder performs better than SigLIP across all setups. Joint scaling of both the diffusion model and the image encoder works best for both setups.

transformer stage. The resolution of the lowest feature map is $16 \times 16$. Transformer blocks operate at the stages with the two lowest resolutions $16 \times 16$ and $32 \times 32$ and convolutional blocks operate in the remaining stages. The four downsampling stages have 128, 128, 256 and 512 channels and the final transformer has 1024 channels. The first three stages have three blocks each and the last stage has sixteen blocks. The optimizer is Adam (Kingma & Ba, 2014) with parameters $\beta_1 = 0.9$, $\beta_2 = 0.99$, $\epsilon = 1e^{-12}$, batch size of 2048 and a learning rate of $2e^{-4}$. We also use Polyak averaging with a decay factor of $0.9999$. The diffusion loss parameters include v-prediction with loss in epsilon phase and a cosine adjusted schedule with a noise resolution of 32. We use the DDPM sampler with an interpolation of 0.2 (standard deviation is $\sigma_{t \to s}^{0.2} \sigma_{st}^{0.8}$) and 0.5 guidance for our ablations. Each ablation run utilizes 256 TPUv3 (Google, 2023) chips around 300K steps. However, the consistent ranking of different ablations throughout training can allow for a much shorter training schedule to identify the best model. We report the FID on 50000 ImageNet samples. Our final *Semantica* model that operates on $512 \times 512$ has a $2 \times 2$ patchification layer instead of $256 \times 256$. We then use 128, 256, 1024, 2048 and 4096 channels with 2, 3, 3, 3 and 12 blocks each.

## 6.2 CHOICE OF IMAGE ENCODER

We first compare two choices of conditioning the diffusion model on frozen image embeddings. Global feature conditioning with FiLM layers and local feature conditioning with cross-attention. Fig. 9 shows that cross-attention with local features, consistently outperforms FiLM across both SigLIP and DINO encoders. The result highlights the importance of local features for image variations.

Then, we investigate the impact of scaling pretrained encoders and diffusion models. We evaluate eight combinations of two encoders each with Base and Large scales and two scales of diffusion models (SiD B and SiD L). Table. 1 reports the FID of each of these combinations at 300K steps. Scaling the encoder while keeping the diffusion model fixed, offers improvements ranging from -0.8 for (SiD-L + DINO) to -1.5 for (SiD-B + SigLIP). Scaling the diffusion model with a fixed encoder size consistently improves FID by around -4.0 for all encoders. Finally, jointly scaling the encoder and diffusion model together results in significant improvements: DINOv2 improves from 13.0 to 9.0 and SigLIP from 17.1 to 11.2. Thus, from here on we will use DINOv2 as the frozen image encoder.

## 7 METRICS FOR IMAGE VARIATIONS

Previous methods assess image variations using the following approach. For each reference image in a test set, the model generates a new image. To quantify the quality of image variations, these methods employ LPIPS (Zhang et al., 2018) for individual image comparisons and FID (Heusel et al., 2017) or Precision/Recall (Kynkäänniemi et al., 2019) to compare test to generated distributions. These metrics can be sufficient to compare variations of our model on our episodic dataset since we explicitly filter out near duplicates, and in theory, the model is unlikely to repeat the same image. However, one major drawback is its inability to measure how diverse the generates samples are with respect to the input image. For instance, a model that just copies the input image or produces very minor variations will have near perfect LPIPS and FID. Thus, these metrics are not ideal for image variation baselines that rely on reconstruction.

To address this limitation, we propose a few-shot approach to assess image variations. Given a set of $N$ test images, we randomly select $K$ images and generate $N/K$ samples for each, resulting in

a total of $N$ samples. We then evaluate FID, recall and precision between the $N$ test images and $N$ samples. Precision measures the quality of the generated samples while recall measures sample diversity.

We provide a short recap on recall a widely used metric to capture diversity between a real and generated dataset. For each generated sample, we store the distance to its Kth nearest neighbor, which serves as a per-sample threshold. A real image has recall 1.0 if its distance to any generated image is less than its corresponding threshold. If a model simply copies a real image, the distance between the original and the copy is zero, which will always be less than its threshold.

Imagine a toy real dataset with 1 class and ten images. To measure 1-shot recall, we condition the image variation model on just one image randomly sampled from the ten images and generate ten samples. If the model just copies the same image ten times, then its nearest neighbour threshold per image is zero. However, the distance between each of the nine other images and the conditioning image is greater than zero, which is greater than the threshold of zero. Thus each of the nine images have a recall of 0.0.

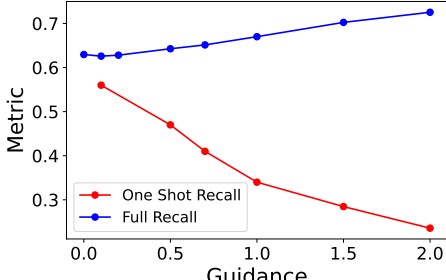

Figure 3: One-shot recall vs full recall on varying guidance. On increasing guidance, thus reducing diversity, one-shot recall decreases while recall on the full dataset counter intuitively increases.

We empirically illustrate this behavior by comparing recall against one-shot recall in Fig. 3. We control the diversity of the image diffusion model by varying the guidance. Note that higher Guidance results in lower diversity but higher precision. A good metric for diversity should therefore drop with higher guidance. This is not the case for full-dataset recall which actually increases with higher Guidance (=lower diversity). Conversely, one-shot recall decreases with higher Guidance (=lower diversity), and is therefore a better metric for diversity.

## 8 COMPARISONS

### 8.1 BASELINES

We compare Semantica to state-of-the-art image variation baselines: Versatile Diffusion (Xu et al., 2023b), Stable Diffusion v2 Image Variations and IP-Adapter (Ye et al., 2023) on generating image variations of size $512 \times 512$. As seen in Sec. 2.1, all baselines rely on image reconstruction to generate image variations. We sweep across a range of guidance factors for all baselines. See App. H) for detailed results of FID with respect to guidance.

### 8.2 IMAGENET ONESHOT

**Setup.** We compare *Semantica* with Versatile Diffusion, IP-Adapter and SD-v2 Image Variations on one-shot ImageNet. Remember that ImageNet has a total of thousand classes. We sample ten images randomly per-class and create a ground truth set of 10000 images. Each baseline model receives one image per-class and generates ten samples per-image with different random seeds, leading to a total of 10000 samples. We then compute FID, precision and recall between 10000 ground truth images and 10000 samples.

**Results.** Fig. 4 left reports the one-shot FID of all four models. For each model, we tune the guidance factors. Table 6 provides detailed results on the relationship between guidance factors

| Model | One-Shot FID |
|---|---|
| SD Image Variations v2 | 34.5 |
| Versatile Diffusion | 26.3 |
| IP-Adapter | 20.2 |
| Semantica | 18.5 |

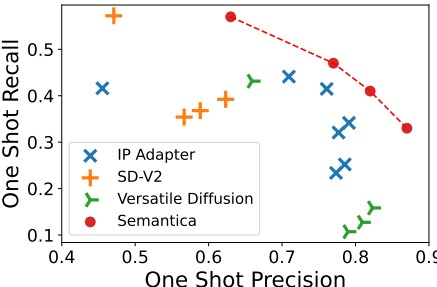

Figure 4: Comparison of *Semantica* against three state-of-the-art image variation baselines on one-shot ImageNet, using evaluation metrics: FID (**Left Table**) and Precision-Recall (**Right Plot:**) as evaluation metrics. Each point in Fig. 3 **Right** represents a different guidance factor. Semantica outperforms image-variation baselines achieving lower FID and a better precision-recall tradeoff.

| Model | Preference Rate |
|---|---|
| IP-Adapter | 43 % |
| Semantica | 57 % |

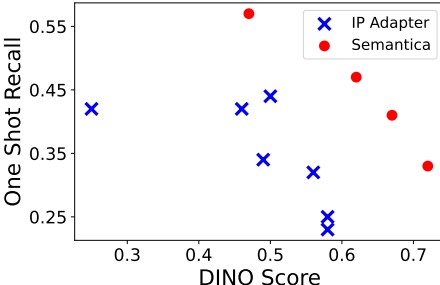

Figure 5: **Left:** We conduct a mechanical turk user experiment to assess the diversity of the models while maintaining consistency with the input image. Semantica achieves a higher user-preference rate as compared to IP-Adapter. **Right:**

and FID for each model. Semantica improves over the previous image variation models achieving a FID of 17.9, 2.3 over the second-best model IP-Adapter. Fig. 4 right displays the precision-recall tradeoff of all models across different guidance factors. Since the conditioning information is present to all models, note that all models have much higher precisions than recall. IP-Adapter and Versatile Diffusion have precision greater than 0.8. SD-V2 IV has higher recall but much lower precision. At lower precisions, Semantica achieves similar recall as compared to SD-V2 IV with much higher precision. At a precision of 0.8, Semantica achieves a high recall higher than the IP-Adpater baseline. Semantica achieves the best-tradeoff between precision and recall among all the baselines.

**User Study.** To assess the diversity of our models while maintaining consistency with the input image, we conduct a user study on Amazon Mechanical Turk. We present the conditioning image and two sets of image randomly selected from either Semantica or IP-Adapter and provide the following prompt. You will see an example image with an object. You get to choose between two alternative sets, Set 1 and Set 2 of related images. Please choose the set that matches the following criteria: 1) The main object of the images in the set should look similar to the example image. 2) There should be diversity between the images in the set. e.g. background and perspective. Semantica demonstrated a significant preference advantage over IP-Adapter, achieving a 57% preference rate compared to 43% for IP-Adapter (95% CI: 54-59%)

**Image Alignment.** We additionally compare alignment between the conditioning and generated image between IP-Adapter and Semantica. We employ two embedding space: CLIP B/16 and DINOv2 B/16 which is known to be better aligned with humans Fu et al. (2023). 5 shows that Semantica achieves a better alignment-recall tradeoff than IP-Adapter in DINO embedding sapce. On CLIP embedding space, Semantica achives slightly better tradeoff or comparable performance to IP-Adapter (See: 11).

|  | ImageNet | Bedroom | Church | SUN397 |
|---|---|---|---|---|
| Label grouped | **4.8** | 46.2 | 27.1 | 29.7 |
| Semantica | 18.4 | **6.2** | **17.3** | **6.7** |
| Guidance @ 0.5 | | | | |
| Label grouped | **5.1** | 34.2 | 20.4 | 22.4 |
| Semantica | 6.2 | **2.4** | **4.0** | **2.5** |

Table 2: Comparison between Semantica and a Label Grouped baseline (LG) trained on ImageNet. The conditioning and target pairs have the same label for LG. LG outperforms Semantica in-distribution and performs worse on out-of-distribution datasets.

### 8.3 LABEL GROUPED BASELINE

Here we compare *Semantica* to a baseline that has direct label supervision (for example, ImageNet) on lower resolution images $256 \times 256$. Recall that the conditioning and target image belong to the same webpage. However, in the presence of label supervision (as in ImageNet), the target and conditioning image can just belong to the same class label. So as a supervised baseline, we group images on ImageNet as per their label and train Semantica on this dataset. Table 2 compares the FID of the Label Grouped baseline (**LG**) to Semantica. Since **LG** is trained on ImageNet, it significantly outperforms Semantica (FID 4.8 vs FID 18.4). However, this trend reverses on all other datasets, where Semantica outperforms **LG**. Both the supervised baseline and Semantica rely on the DINOv2 encoder which was trained on a wide variety of data sources. Therefore the encoder itself may provide useful representations on a number of datasets. But training **LG** just on ImageNet, might limit the diffusion model's exposure from non ImageNet images, potentially explaining its significant performance drop on all other datasets.

## 9 CONCLUSION AND LIMITATIONS

Our paper explores a new method for training image-conditioned diffusion models to generate image variations. Instead of the typical image reconstruction approach, we condition the model on one random image from a webpage and train it to denoise another random image from the same webpage. Through rigorous evaluations, DINOv2 as the image encoder produces better image variations than the popular CLIP model. Finally, we emphasize the difficulty in measuring image variations, and propose new metrics that are applicable in the one-shot setting.

In this work, we focus on evaluating image variations on datasets consisting of mainly a single object. When multiple objects are present in an image, additional supervision in the form of bounding boxes or text can allow for fine-grained control of generations. Further, as in prior works, we focus on frozen image encoders to efficiently encode representations and filter data as opposed to training an image encoder end-to-end. Thus *Semantica* can inherit the biases of the frozen image encoder. We leave studying the tradeoffs between finetuning and using frozen representations to future work.

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

# A  MORE SAMPLES: IMAGENET

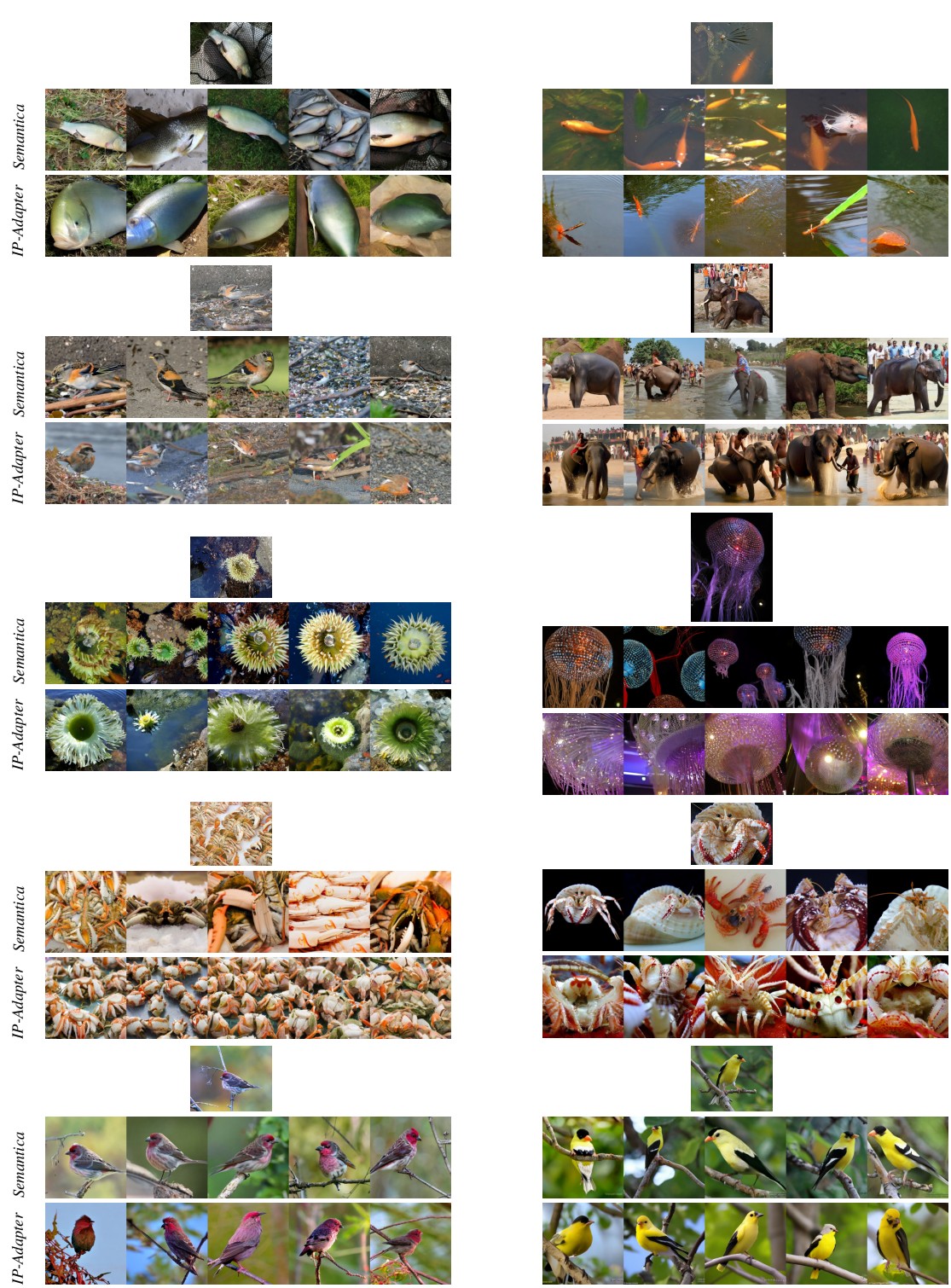

Figure 6: We present additional samples and comparisons on ImageNet. Samples from *Semantica* reflect diversity while being congruent with the conditioning image.

## B    CLIP: RECONSTRUCT IMAGES

We train a diffusion model conditioned on SigLIP embeddings to reconstruct the original image. Fig. 7 shows four samples images from the ImageNet validation set and the corresponding generations from the generative model.

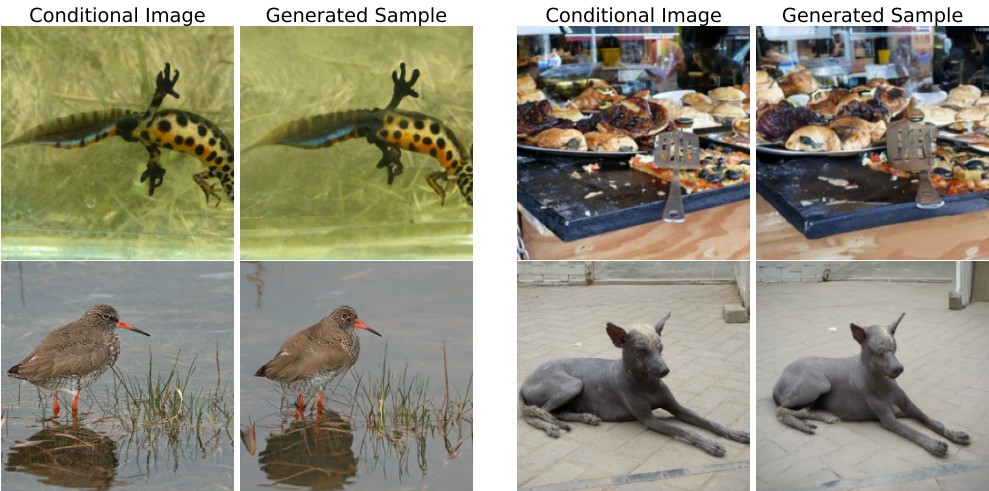

Figure 7: A conditional diffusion model reconstructs images from frozen SigLIP embeddings. As seen in the case with frozen DINOv2 embeddings in Fig.2, the generated samples exhibit very minor low-level variations.

## C    DATA FILTERING

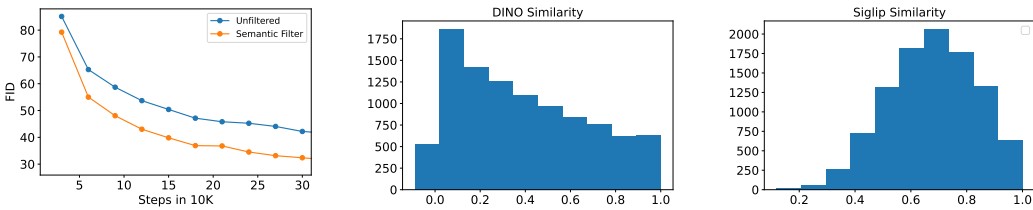

Figure 8: **Left:** FID of DINO-v2 B/14 + Cross Attention with and without data filtering. **Center:** Histogram of DINO Similarities between Episodic WebLi image pairs. **Right:** Histogram of SigLIP Similarities between Episodic WebLi image pairs

Here, we investigate the impact of semantic data filtering. We first manually looked at pairs of images from the Episodic Webli training set and computed their similarites in DINO embedding space. We found a lower threshold of 0.3 to be sufficient to filter out completely unrelated images and 0.9 to filter out near duplicates. Interestingly. we also found that the distribution of similiarities to be dependent on the embedding space used. For example, DINOv2 (Fig. 8 Center) assigns more examples a lower similarity as compared to SigLIP (Fig. 8 Right). So we set the lower threshold of CLIP and DINOv2 models such that, the total number of examples are roughly the same. This lead to a lower threshold of 0.65 for CLIP. Fig. 8 middle shows the FID of the DINO-v2 B/14 + Cross Attention with and without data filtering. Similarity-based data filtering in DINO feature space positively impacts the generation quality and improves FID by greater than 10. In future work, we can explore tuning these thresholds for a desired quality-diversity tradeoff or even directly conditioning the diffusion model on the desired similarity with the conditioning image.

## D  FILM VS CROSS-ATTENTION

Here, we compare Film based conditioning to cross-attention based conditioning.

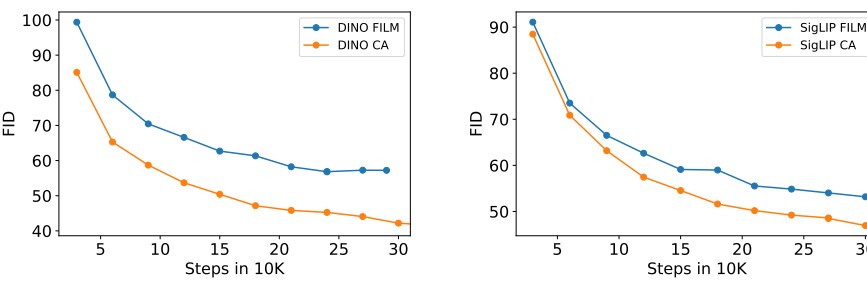

Figure 9: We plot ImageNet FID as a function of number of training steps on Episodic WebLI. **Left:** DINO-v2 B/14 with Film and cross attention **Right:** SigLIP B/14 with film and cross attention.

## E  MAE ENCODER

We also experiment with a frozen MAE Enocder. Plugging in a ViT-L MAE encoder has reasonable results on one-shot FID but performs slightly worse than SigLIP ViT-L. Fig. 10 compares the one-shot FID of DINO-v2, SigLIP and MAE with ViT-L Image Encoders.

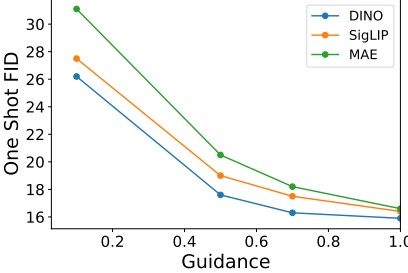

Figure 10: We compare three models: DINO-v2, SigLIP and MAE with a ViT-L image encoder on one-shot ImageNet FID.

## F  CLIP ALIGNMENT

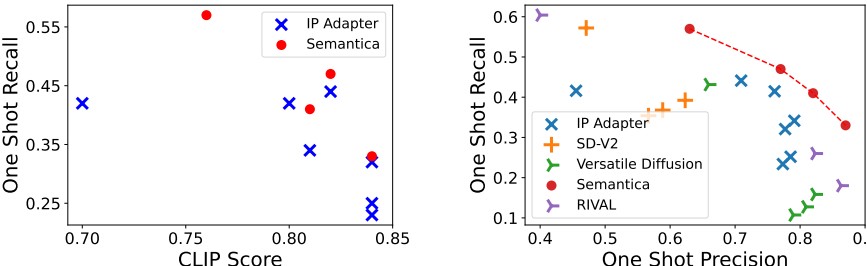

Figure 11: **Left:** Semantica achives slightly better tradeoff or comparable performance to IP-Adapter on CLIP Alignment. **Right:**

## G  COMPARISON WITH RIVAL

RIVAL employs additional text-based conditioning and therefore does not form a direct baseline to our model. Nevertheless, we compared Semantica to RIVAL with text conditioning based on imagenet class names. On one-shot FID, RIVAL achieves a FID of 17.5 and outperforms Semantica by 1 FID point. However, Fig. 11 demonstrates that RIVAL achieves a worse precision-recall tradeoff.

## H  IMAGENET ONE-SHOT FID VS GUIDANCE HYPERPARAMETERS

We report fine-grained FID results for different guidance values. Tab. 3 reports fine-grained FID results for SD-v2 IV and Versatile Diffusion. Tab. 5, reports results for IP-Adapter and Semantica.

| Guidance | FID | | Guidance | FID |
|---|---|---|---|---|
| 1.0 | 46.7 | | 1.0 | 28.5 |
| 4.0 | **30.8** | | 4.0 | **26.3** |
| 6.0 | 34.5 | | 6.0 | 29.4 |
| 8.0 | 37.6 | | 8.0 | 31.8 |

Table 3: Guidance against one-shot ImageNet FID. **Left:** SD-IV and **Right:** Versatile Diffusion

| Guidance | Scale | FID |
|---|---|---|
| 1.0 | 0.5 | 57.4 |
| 4.0 | 0.5 | 25.4 |
| 7.0 | 0.5 | 24.5 |
| 1.0 | 1.0 | 42.6 |
| 2.0 | 1.0 | 23.2 |
| 4.0 | 1.0 | **20.2** |
| 7.0 | 1.0 | 20.6 |
| 7.0 | 1.0 | 21.2 |

| Guidance | FID |
|---|---|
| 0.1 | 29.4 |
| 0.5 | 21.0 |
| 1.0 | 18.5 |

Table 4: Semantica

Table 5: Guidance against one-shot ImageNet FID. **Left:** IP Adapter and **Right:** Semantica

## I  SUN-397 ONESHOT

This experiment compares *Semantica* with IP-Adapter on one-shot SUN-397. SUN-397 has a total of 397 classes. We sample 25 images randomly per-class and create a ground truth set of 9925 images. Each model generates 25 samples given a randomly sampled image, leading to total of 9925 samples. Similar to ImageNet, Fig. 12 reports FID, precision and recall between the 9925 generated samples and ground-truth images. *Semantica* achieves a one-shot FID of 13.0, outperforming IP-Adapter. It also achieves a much more favourable precision-recall tradeoff.

| Guidance | Scale | FID |
|---|---|---|
| 1.0 | 0.5 | 24.6 |
| 4.0 | 0.5 | **14.1** |
| 7.0 | 0.5 | 16.7 |
| 1.0 | 1.0 | 20.0 |
| 4.0 | 1.0 | 17.8 |
| 7.0 | 1.0 | 27.9 |

| Guidance | FID |
|---|---|
| 0.1 | 13.2 |
| 0.3 | 12.6 |
| 0.5 | 13.0 |
| 0.7 | 13.5 |
| 1.0 | 14.7 |

Table 6: Guidance against one-shot SUN397 FID. **Left:** IP Adapter and **Right:** Semantica

| Model | One-Shot FID |
|---|---|
| IP-Adapter | 12.8 |
| Semantica | 12.6 |

Figure 12: Comparison of *Semantica* against IP-Adapter on one-shot SUN397, using evaluation metrics: FID (**Left Table**) and Precision-Recall (**Right Plot:**) as evaluation metrics.

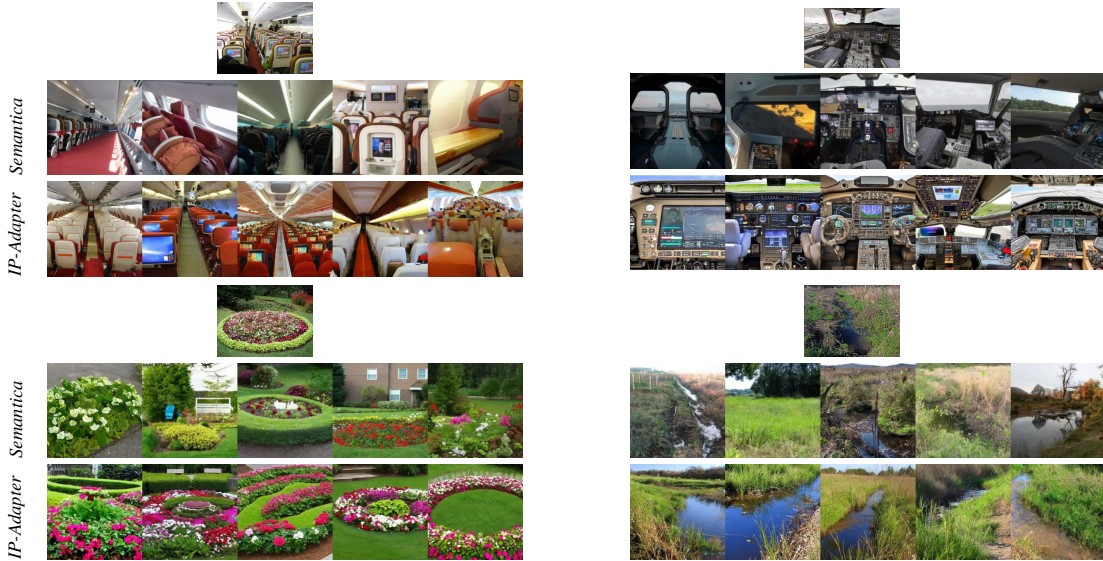

Figure 13: We present additional samples and comparisons on SUN397

## J  QUALITATIVE EFFECT OF GUIDANCE

Fig. 14, displays five conditioning images from ImageNet and the generated samples at different guidance factors. At guidance factor 0.0, the samples reflect a broad semantic category from the conditioning image. Increasing the guidance factor leads to samples that incorporate more specific details from the conditioning image. For example, with the conditioning image of the dog and the kid, Semantica stars with a sample of a dog. The specific breed of the dog and the child in the image appear as we amplify the guidance. Fig. 15 showcases samples for each small dataset across various guidance factors. In row four, the bed (main object) persists across all guidance levels, while the chair and the fence appear at high guidance levels. Row five exhibits a similar effect: the number of minarets in the generated church increases from one to two and the shape of the main dome begins to resemble the conditioning image. In row one, the sample resembles toys with zero guidance, the sample resembles toys but transforms into a crowded convention as guidance increases.

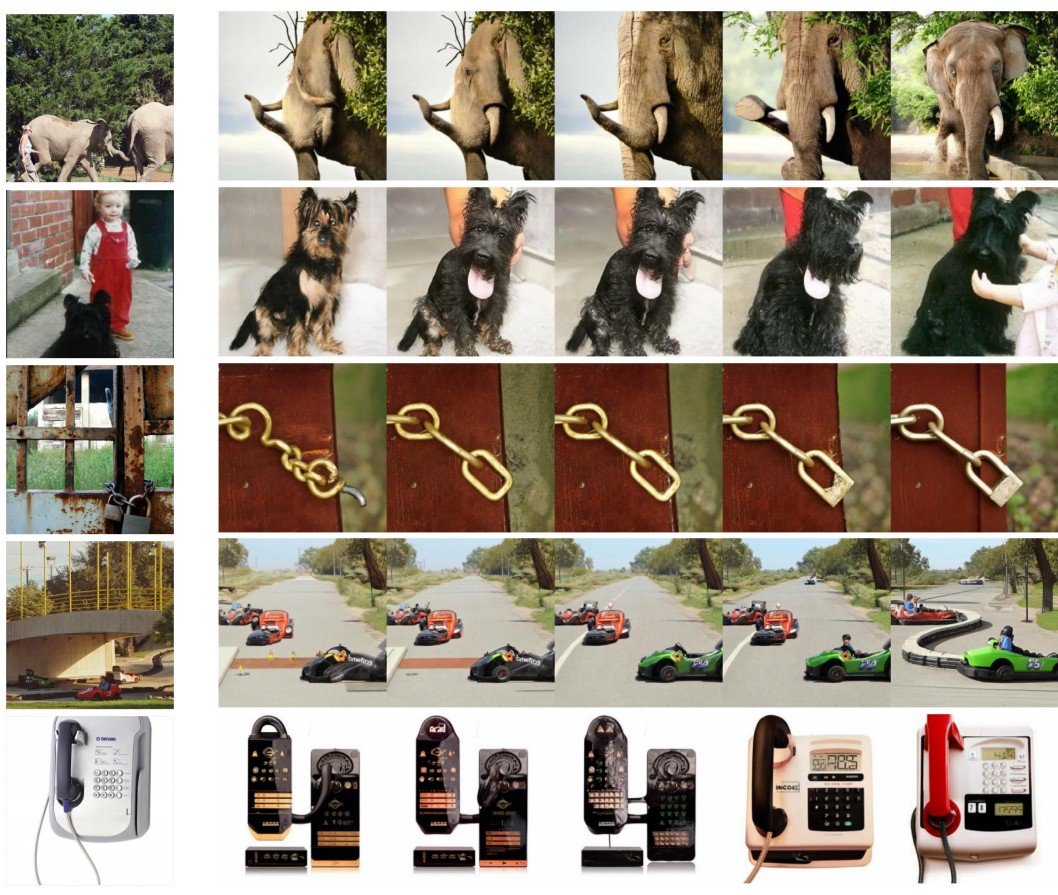

Figure 14: **Left:** Conditioning Image from ImageNet. **Right:** Generated samples with guidance factors 0.0, 0.1, 0.2, 0.5 and 1.0. At guidance factor 0.0, the samples reflect a broad semantic category from the conditioning image. Increasing the guidance factor leads to samples that incorporate more specific details from the conditioning image.

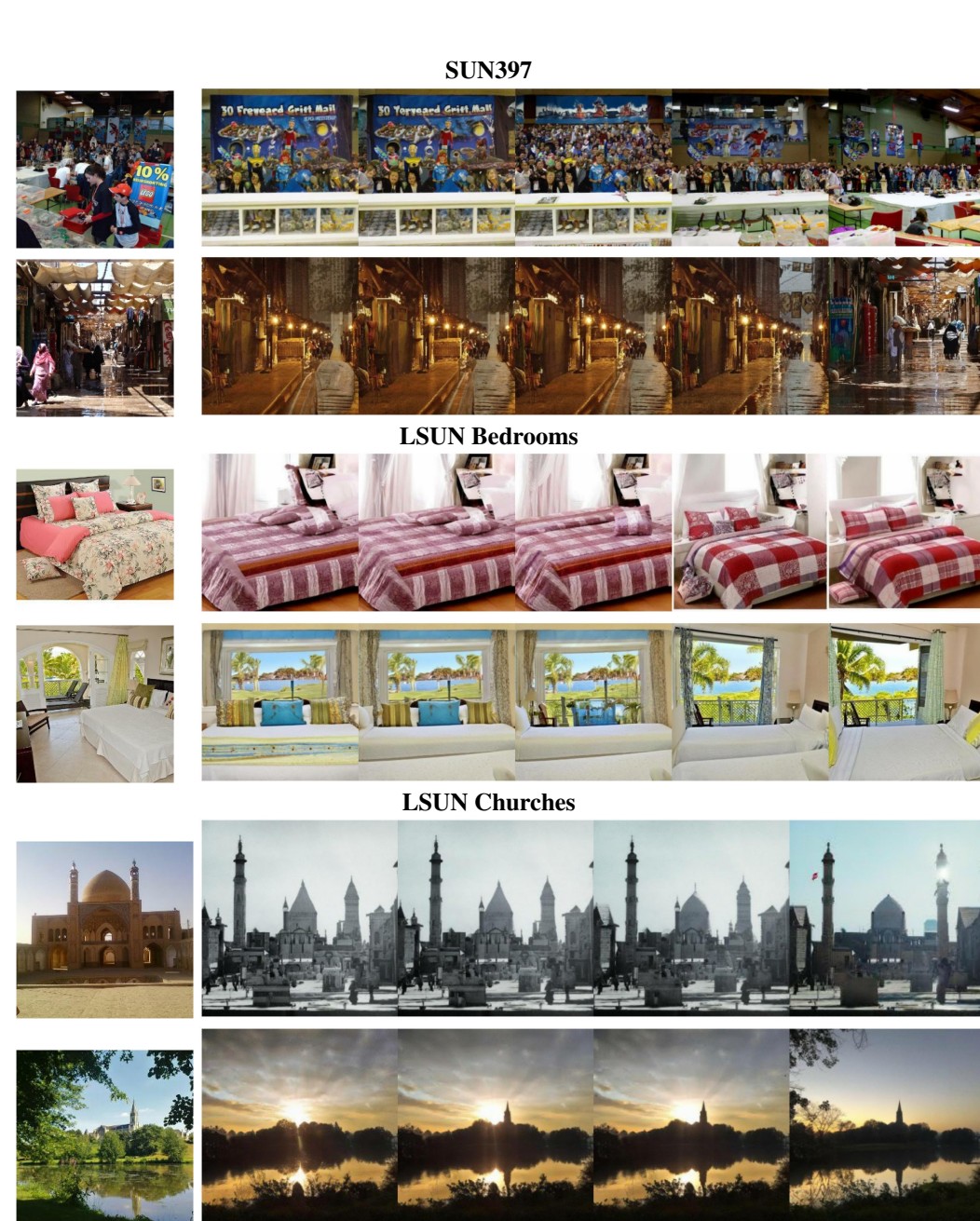

Figure 15: **Left:** Conditioning Images from *SUN397* (Top two rows), *LSUN Bedrooms* (Middle two rows) and *LSUN churches* (Last two rows). **Right:** Generated samples with guidance factors 0.0, 0.1, 0.2, 0.5, 1.0 and 1.5.

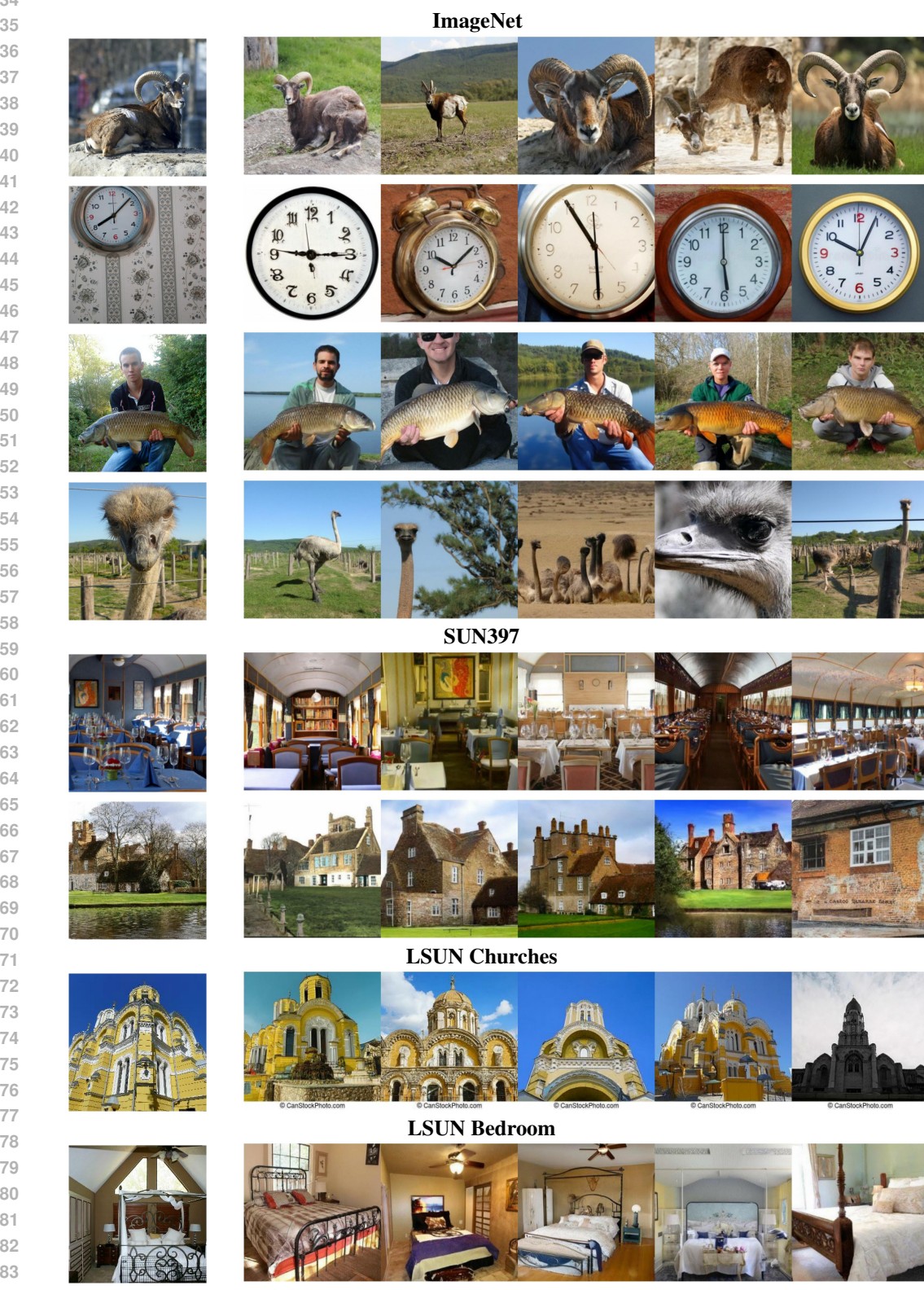

Figure 16: **Left:** Conditioning image **Right:** Five samples at guidance 0.5. Semantica is trained exclusively on web-image pairs. During adaptation, it receives a conditioning image and generates samples reflective of semantic information. Semantica requires no label supervision or finetuning.

