# OpenReview forum: "Conditional Diffusion on Web-Scale Image Pairs leads to Diverse Image Variations"
_ICLR.cc/2025/Conference — Submitted to ICLR 2025_

### Official Review · Reviewer_n2nW · 2024-10-31

**Soundness:** 3
**Presentation:** 3
**Contribution:** 3
**Rating:** 6
**Confidence:** 3

**Summary:**

The paper if focus on the challenge of generating image variations, with large diversity while preserving the the conditions' semantics.
The authors suggest a new pre-training method exploiting the semantic relations of images within the same web page.
It also discuss the challenge of measuring image diversity and suggest a few-short metrics approach

**Strengths:**

- The idea of using the semantic relations of image pairs with in the the same web page is a good contribution which can lead to even more  research direction then the focus of this paper
- The observation of image diversity issue in current SOTA , mitigate it and suggest new metric to evaluate it

**Weaknesses:**

The claim that "Standard image-level metrics such as LPIPS and distribution-level metrics such as FID fail
to capture diversity in image variations" is not backup with number and/or examples

**Questions:**

Please provide specific examples or quantitative evidence demonstrating how LPIPS and FID fail to capture diversity in image variations and should do a comparative analysis between the proposed one-shot metrics and LPIPS/FID, including specific examples where the new metrics better capture diversity.

---

> ### Author Response · Authors · 2024-11-25
> **Rebuttal Response**
>
> Thanks for the review! In response to your suggestion, we further expanded Section 7 (marked in red)
>
> The challenge in evaluating diversity is that many quality metrics can be maximized by simply copying the image, which would obviously provide low diversity. This also includes full-dataset recall which has been invented specifically to evaluate diversity of generative models.
>
> We provide a short recap on recall a widely used metric to capture diversity between a real and generated dataset. For each generated sample, we store the distance to its Kth nearest neighbor, which serves as a per-sample threshold. A real image has recall 1.0 if its distance to any generated image is less than its stored threshold. If a model simply copies a real image, the distance between the original and the copy is zero, which will always be less than its threshold.
>
> Imagine a toy real dataset with 1 class and ten images. To measure 1-shot recall, we condition the image variation model on just one image randomly sampled from the ten images and generate ten samples. If the model just copies the same image ten times, then its nearest neighbour threshold per image is zero. However, the distance between each of the nine other images and the conditioning image is greater than zero, which is greater than the threshold of zero. Thus each of the nine images have a recall of 0.0 which emphasizes the poor diversity of the image variation model.
>
> We empirically illustrate this behavior by comparing recall against one-shot recall in Fig.3. We control the diversity of the image diffusion model by varying the guidance. Note that higher Guidance results in lower diversity. A good metric for diversity should therefore drop with higher guidance. This is not the case for full-dataset recall which actually increases with higher Guidance (=lower diversity). Conversely, one-shot recall decreases with higher Guidance (=lower diversity), and is therefore a better metric for diversity.

---

### Official Review · Reviewer_2azc · 2024-11-03

**Soundness:** 2
**Presentation:** 2
**Contribution:** 3
**Rating:** 5
**Confidence:** 3

**Summary:**

This paper works on image variations generation based on two main observations. Firstly, conditional diffusion models with frozen condition embeddings can reconstruct the image with minor variations. Secondly, web-scale image can be used to train image variation models. This paper then examine various design choices for condition representation generation given its crucial role in the proposed method, and find DINOV2 can provide good image representation for better image variation generation.

**Strengths:**

1. With web-scale images, this paper successfully find a good application on top of it to perform Image variation generation, which is inspiring and effective as illustrated in the experiments section.
2. The final conclusion that DINOV2 produces image representation for better image variations is also interesting, explaining self-supervised representation learning is promising in achieving effective image generation/editing.
3. The proposed new way of evaluate performance of image variation techniques seems interesting and inspiring.

**Weaknesses:**

1. The analysis on image encoder part for image variation (sec 2.2) seems not informative enough, although it's one main contribution of the paper, as it's common sense that the effectiveness of conditional representation is crucial for image generation.
2. A new application of web-scale image for image variation is not well explained. It's not clear how the authors come up with this idea of using web-scale image for image variation, making it hard to evaluate the significant of the contrition.
3. Data filtering should be clearly introduced as it's one of the main contribution. However, Sec. 5 is not clear (see Questions 3).
4. The experimental results section seems weak in proving superiority of the proposed solution. (see Questions 4)

**Questions:**

1. Image representation learning, e.g. self-supervised, image reconstruction based, contrastive learning and etc, are wildly studied. How the other encoders perform compared with DINO? e.g. MAE (ref 1)
2. Web-scale image is indeed Episodic WebLI in this paper. Are there any other web-scale images? Why choose Episodic WebLI in particular?
3. Please explain robustness of the method with respect to the cleanest of the web-scale images (line 306-315), e.g. what if the web-scale images are not well filtered out, how the method perform then?
4. As explained in the paper, the commonly used metrics, e.g FID, fail to evaluate variation performance. However, it's still not clear how the proposed solution (sec 7) can be effective in measure diversity, e.g. Fig. 3 should be further analysed to explain diversity superiority.

re1: Masked Autoencoders Are Scalable Vision Learners, CVPR 2022

---

> ### Author Response · Authors · 2024-11-25
> **Rebuttal Response**
>
> Thanks for the review! We revised the paper with significant changes in response to the review and all changes are marked in red.
>
> --------
>
> ## MAE Baseline
> We also experiment with a frozen MAE Enocder. Plugging in a ViT-L MAE encoder has reasonable results on one-shot FID but performs
> slightly worse than SigLIP ViT-L. Fig. 9 and App. E compares the one-shot FID of DINO-v2, SigLIP and MAE with ViT-L Image Encoders.
>
> ----------------------
>
> ## Episodic WebLI
> Vision Language models trained on Episodic Webli, have shown to have excellent performance on VLM tasks. Thus it forms an attractive choice also for image generative modeling. The closest alternative to episodic webli is multimodal C4. Unfortunately, the raw images are not made public, thus rendering it unsuitable for image modeling.
>
> ---------------------
>
> ## Data Filtering
>
> We found that filtering the training data was crucial to obtain good image variation results. In our initial experiments, we did not do this and qualitatively, it resulted in two failure modes: Generation of near duplicates and generation of unrelated images. Figure 7 of our original submission quantitatively shows that training on unfiltered data resulted in a significant decrease in performance (around 10 FID points). This highlights the importance of our filtering approach.
>
> Our filtering process leverages the same image encoder used in our image diffusion model. For instance, if we use SigLIP to encode images for generation, we also use it to filter the data.
>
> Finally, we added more clarity in App. C on how the lower and higher thresholds were set for different models. "We first manually looked at pairs of images from the Episodic Webli training set and computed their similarites in DINO embedding space. We found a lower threshold of 0.3 to be sufficient to filter out completely unrelated images and 0.9 to filter out near duplicates. Interestingly. we also found that the distribution of similiarities to be dependent on the embedding space used. For example, DINOv2 (Fig. 7 Center) assigns more examples a lower similarity as compared to SigLIP (Fig. 7 Right). So we set the lower threshold of CLIP and DINOv2 models such that, the total number of examples are roughly the same. This lead to a lower threshold of 0.65 for CLIP."
>
> ---------------------
>
> ## One-shot Recall vs Full Recall on diversity
>
> We expanded Section 7 to provide more intuition and emprical evidence comparing one-shot Recall to full Recall.
>
> The challenge in evaluating diversity is that many quality metrics can be maximized by simply copying the image, which would obviously provide low diversity. This also includes full-dataset recall which has been invented specifically to evaluate diversity of generative models.
>
> We provide a short recap on recall a widely used metric to capture diversity between a real and generated dataset. For each generated sample, we store the distance to its Kth nearest neighbor, which serves as a per-sample threshold. A real image has recall 1.0 if its distance to any generated image is less than its stored threshold. If a model simply copies a real image, the distance between the original and the copy is zero, which will always be less than its threshold.
>
> Imagine a toy real dataset with 1 class and ten images. To measure 1-shot recall, we condition the image variation model on just one image randomly sampled from the ten images and generate ten samples. If the model just copies the same image ten times, then its nearest neighbour threshold per image is zero. However, the distance between each of the nine other images and the conditioning image is greater than zero, which is greater than the threshold of zero. Thus each of the nine images have a recall of 0.0 which emphasizes the poor diversity of the image variation model.
>
> We empirically illustrate this behavior by comparing recall against one-shot recall in Fig.3. We control the diversity of the image diffusion model by varying the guidance. Note that higher Guidance results in lower diversity. A good metric for diversity should therefore drop with higher guidance. This is not the case for full-dataset recall which actually increases with higher Guidance (=lower diversity). Conversely, one-shot recall decreases with higher Guidance (=lower diversity), and is therefore a better metric for diversity.

---

> > ### Author Response · Authors · 2024-12-02
> >
> > Since the deadline is approaching, please let us know if you have further questions.
> > A summary of our changes can be found here (https://openreview.net/forum?id=s7vwXDsVYa&noteId=Y2uvvhPQei)

---

### Official Review · Reviewer_1V2G · 2024-11-04

**Soundness:** 2
**Presentation:** 4
**Contribution:** 2
**Rating:** 5
**Confidence:** 3

**Summary:**

This paper introduces a new model, Semantica, which generates image variations given contextual input images. The model is trained on web-scale image pairs. Specifically, it is trained with pairs of images from the same webpage. Multiple design choices are discussed in the paper, such as the selection of the image encoder, diffusion decoder, etc. The model is evaluated on a few-shot version of FID, recall, and precision for the generation of image variations and outperforms prior works such as SD-v2, IP-adapter, and Versatile Diffusion.

**Strengths:**

* Generating diverse image variations by training on images from the same webpage is both simple and interesting.
* The paper is well-written and easy to follow. The visualizations seem to outperform the prior works listed in the paper.
* The designs and ablation studies of different model components are well explained.

**Weaknesses:**

* There is a missing comparison with some related work, such as RIVAL [1] which also targets the image variation task.
* Few-shot FID, precision, and recall may not be sufficient to fully quantify performance. Some metrics proposed in the table 1 of RIVAL [1] could provide more insightful for evaluations.
* There is a lack of a user study to compare different methods, which would be more convincing, as numerical metrics may not fully quantify performance.


**Reference**

[1] Real-World Image Variation by Aligning Diffusion Inversion Chain, NeurIPS 23

**Questions:**

1. In eqn 1, why there is 2 variable t? One of them is context?

---

> ### Author Response · Authors · 2024-11-27
> **Rebuttal Response**
>
> Thanks for the review! We revised the paper with significant changes in response to the review and all changes are marked in red.
>
> ---
>
> ## User Study
>
> To assess the diversity of our models while maintaining consistency with the input image, we conduct a user study on Amazon Mechanical Turk. We present the conditioning image and two sets of image randomly selected from either Semantica or IP-Adapater and provide the following prompt.
>
> "You will see an example image with an object. You get to choose between two alternative sets, Set 1 and Set 2 of related images. Please choose the set that matches the following criteria: 1) The main object of the images in the set should look similar to the example image. 2) There should be diversity between the images in the set. e.g. background and perspective."
>
> **Semantica demonstrated a significant preference advantage over IP-Adapter, achieving a 57 % preference rate compared to 43\% for IP-Adapter (95% CI: 54-59%)**. Note that this closely follows our quantitative results in Fig. 4. We expanded Section 8.2 with these results
>
> ---
>
> ## RIVAL Comparison
>
> RIVAL employs additional text-based conditioning as compared to Semantica (and other baselines) that emply only image-based conditoning. (See: page 6 of RIVAL, “In contrast, RIVAL generates image variations based on text descriptions and reference images, harnessing the real-image inversion chain to facilitate latent distribution alignments” and the provided code (https://github.com/dvlab-research/RIVAL/blob/main/rival/test_variation_sdv1.py#L89). Therefore RIVAL does not form a direct baseline to our model.
>
> Nevertheless, we compared Semantica to RIVAL with text conditioning based on imagenet class names. On one-shot FID, RIVAL achieves a FID of 17.5 and outperforms Semantica by 1 FID point. However, Fig. 11 demonstrates that RIVAL achieves a worse precision-recall tradeoff. Overall, Semantica is competitive with a model that incorporates both text + image conditioning. We added these new results to App. G
>
> ---
>
> ## Image Alignment
>
> We additionally include image alignment between the conditioning and generated images. We compare IP-Adapter and Semantica on two different embedding spaces CLIP B/16 and DINOv2 B/16. Note that similarity as computed in DINO embedding space has shown to be better aligned with human perception as compared to CLIP Embeddings (Ref: Fig 4 in https://arxiv.org/pdf/2306.09344). Fig. 5 shows that Semantica achieves a better alignment-recall tradeoff than IP-Adapter in DINO embedding sapce. On CLIP embedding space, Semantica achives slightly better tradeoff or comparable performance to IP-Adapter (See:Fig. 11). We expanded Section 8.2 with these results.
>
> ---
>
> In eqn 1, why there is 2 variable t? One of them is context? -> Yes, there should be just one. we will fix this typo.

---

> > ### Author Response · Authors · 2024-12-02
> >
> > Since the deadline is approaching, please let us know if you have further questions. A summary of our changes can be found here (https://openreview.net/forum?id=s7vwXDsVYa&noteId=Y2uvvhPQei)

---

### Official Review · Reviewer_kdXF · 2024-11-04

**Soundness:** 3
**Presentation:** 2
**Contribution:** 2
**Rating:** 3
**Confidence:** 5

**Summary:**

This paper focuses on the task of generating image variations and proposes the Semantica model. Unlike traditional diffusion model training, Semantica is trained on web-scale image pairs. It receives a random (encoded) image from a webpage as conditional input and denoises another noisy random image from the same webpage. Additionally, the paper explores the impact of different image encoders on the results.

**Strengths:**

1. This paper is easy to understand.
2. The proposed method achieves good results on several benchmarks.

**Weaknesses:**

1. I have some doubts about the task of generating image variations. The model trained in this paper only supports image input and does not support text guidance, so it doesn’t seem to have strong practical value for real-world image editing. Additionally, many AIGC models already support various tasks such as image editing and style transfer. Therefore, I don't quite understand the value of the generating image variations task.
2. Emu2 [1] has already discovered that using CLIP image embeddings as conditional inputs for training a diffusion model can successfully reconstruct images, which is similar to the findings in this paper.
3. From the visual results, it is difficult to see any advantage of the proposed method over IP-Adapter. Moreover, IP-Adapter has broader application scenarios, as it can adapt to various text-to-image models.



[1] Generative Multimodal Models are In-Context Learners. CVPR, 2024.

**Questions:**

Refer to Weakness.

---

> ### Author Response · Authors · 2024-11-27
> **Rebuttal Response**
>
> Thanks for the review! We revised the paper with significant changes in response to the review and all changes are marked in red.
>
> ----
>
> ## User Study
>
>
>
> In addition to the presented quantitative results (Fig 4), we further showcase the advantage of Semantica over IP-Adapter via a user study. To assess the diversity of our models while maintaining consistency with the input image, we conduct a user study on Amazon Mechanical Turk. We present the conditioning image and two sets of image randomly selected from either Semantica or IP-Adapater and provide the following prompt.
>
> "You will see an example image with an object. You get to choose between two alternative sets, Set 1 and Set 2 of related images. Please choose the set that matches the following criteria: 1) The main object of the images in the set should look similar to the example image. 2) There should be diversity between the images in the set. e.g. background and perspective."
>
> Semantica demonstrated a significant preference advantage over IP-Adapter, achieving a 57 % preference rate compared to 43% for IP-Adapter (95% CI: 54-59%). Note that this closely follows our quantitative results in Fig. 4. We expanded Section 8.2 with these results
>
> ----
>
>
> ## Emu2
>
> Thank you for bringing Emu2 to our attention; we will certainly cite it. However, our work fundamentally differs from Emu2 in both motivation and contribution. While Emu2 achieves superior image reconstruction quality using CLIP embeddings, our primary objective is to disentangle the source of variation in text-to-image diffusion models.
>
> We first establish a baseline: a vanilla diffusion model trained solely on image reconstruction. This baseline, unlike existing models trained with auxiliary tasks, demonstrates a limited capacity for generating variations. This key observation leads us to hypothesize that the rich image variations observed in existing models stem from the co-training process itself, rather than being solely attributed to the conditional image embeddings.
>
> Finally, we demonstrate that this phenomenon extends beyond CLIP embeddings to other visual representations like DINO embeddings.
>
> ----
>
> ## Purpose of image variations
>
>
> The task of generating image variations holds significant practical value, even without explicit text guidance. Consider dataset augmentation in domains where obtaining diverse, labeled data is difficult or expensive. A good image variation model can create diverse samples from a single image particularly in scenarios where precise textual prompt engineering is difficult.
>
> Secondly, our findings extend beyond the final Semantica model that achieves solid image variation performance. Currently the prevalent recipe is to use CLIP embeddings for image conditioning, however we show the effectiveness of self-supervised embedding like DINO, which could be a promising alternative for image conditioning.
>
> Our work further highlights the limitations of current metrics for image variations; a model that copies the input image can achieve a perfect score. Fig 3 shows that one-shot recall shows more intuitive behaviour with respect to diversity as compared to standard recall metrics used to evaluate generative models.
>
> Finally, we investigate a research question: what enables image variation models to generate diverse outputs, even though they are trained to simply reconstruct the input image? Our experiments suggest that the embeddings are not the source of variation but rather the combined effect of training on multiple auxiliary tasks.
>
> ----

---

> > ### Author Response · Authors · 2024-12-02
> >
> > Since the deadline is approaching, please let us know if you have further questions. A summary of our changes can be found here (https://openreview.net/forum?id=s7vwXDsVYa&noteId=Y2uvvhPQei)

---

### Author Response · Authors · 2024-11-27
**Rebuttal Response**

Thanks to the reviewers! Here is a summary of the new experiments. All the highlighted changes are marked in red.

1. User Study comparing Semantica and IP-Adapter: (Section 8.2 and Figure 5)
2. Comparison of recall and one-shot Recall as diversity is varied. (Figure 3)
3. Image Alignment results:  (Section 8.2 and Figure 5)
4. Comparisons with RIVAL. (App. G)
5. MAE Encoder. (App, E)
6. More clarity on data filtering (App, C and Fig. 8)


We hope our findings on image variations as a whole will be valuable to the ICLR audicence. Please reconsider your scores after the rebuttal revision.

---

### Meta-Review · Area_Chair_n4r1 · 2024-12-18

**Metareview:**

This work targets on generating image variations by training the model with web-scale image pairs. The idea of leveraging the semantic relations of image pairs and simplicity of method are appreciated. However, several main concerns raised by reviewers include the unclear motivation and lack of comparison. It received one clear reject, two borderline reject and one borderline accept. The rebuttal partly addressed some concerns but did not convince reviewers. While this work shows the capability of generating variations of a given image, it didn't present its bigger value beyond variation, which sort of limits the scope of this work. The current evaluation is still insufficient to support claims as well. Considering all the comments and discussions, AC made the decision of reject. The direction is promising but current presentation is incomplete and requires more revision before acceptance. Authors are encouraged to polish it by incorporating all the suggestions from reviewers and consider resubmission elsewhere.

**Additional Comments On Reviewer Discussion:**

The main concerns raised by reviewers include the unclear motivation and lack of comparison. After the rebuttal, one reviewer increase the score from 3 to 5 and another reviewer downgrade from 8 to 6. The other two remain the original score. The reviewer who gave 8 has low confidence and the review is short without enough justifications, thus AC did not put much weight on his/her points. Thus the overall feedback is negative and reviewers unanimously agree that the evaluation is far from being sufficient to support the claim. Therefore the decision of reject is made.

---

### Decision · Program_Chairs · 2025-01-22

Reject